

# Numerical study of hydrodynamic and salinity transport process in Pink Beach wetlands of Liao River Estuary, China

Huiting Qiao[1], Mingliang Zhang[1*], Hengzhi Jiang[2], Tianping Xu[1] and Hongxing Zhang[1]

[1] School of Ocean Science and Environment, Dalian Ocean University, Dalian, Liaoning, 116023, China;

[2] National Marine Environment Monitoring Center, Dalian, Liaoning, 116023, China

*Correspondence*: Mingliang Zhang (zhmliang_mail@126.com)

**Abstract.** The interaction study of vegetation with the flow environment is essential for the determination of the bank protection, morphological characteristics and ecological conditions for the wetlands. This paper uses MIKE 21 hydrodynamic and salinity model to simulate the hydrodynamic characteristics and salinity transport process in Pink Beach wetlands of Liao River estuary. The effect of wetland plant on tidal flow in areas of wetland waters is represented by a varying Manning's coefficient in the bottom friction term. Acquisition of vegetation distribution is based on Landsat TM satellites through remote sensing techniques. The detailed comparisons between field observation and simulated result of water depth, salinity and tidal currents at neap tide and spring tide are presented in vegetated domain of Pink Beach. Satisfactory results are obtained in simulating both flow characteristic and salinity concentration with or without vegetation. Several stations from upstream to downstream in the Pink Beach are selected to estimate the longitudinal variation of salinity under different river runoffs and the results show that the salinity concentration decreases with an increase of river runoff. This study can help to increase understanding of the favorable salinity conditions for the special vegetation growth in the Pink Beach wetlands of Liao River estuary. The results provide crucial guidance for related interaction studies among vegetation, flow and salinity in other wetland waters.

# 1 Introduction

Wetland is a transitional zone between terrestrial ecosystems and aquatic ecosystems, which has a variety of unique functions of providing large amounts of food, raw materials and water resources for humans, maintaining the ecological balance, biodiversity and rare species resources. The aquatic plants in coastal protection from extreme events have become a

recurring question along with the viability assessment of ecosystem-based management approaches (Barbier et al., 2008; Temmerman et al., 2013). Coastal wetlands are mainly distributed in coastal areas in China within eleven areas (Hebei, Liaoning, Shandong, Jiangsu, Zhejiang, Fujian, Guangdong, Hainan, Taiwan, Tianjin and Guangxi). Coastal wetlands cover an area of $5.7959 \times 10^6$ ha, accounting for 10.85% of the total area of wetlands in China, there are 12 types of coastal wetland plants (Jiang et al., 2015). Liao River estuary wetland is located in Panjin City, Liaoning Province, China, with a total

wetland area of about 451300.5 ha (Zhang et al., 2009). The main wetland vegetations include *Phragmites communis* and *Suaeda heteroptera*, which lie in the inter-tidal zone of the Liao River estuary. *Suaeda heteroptera* is a dominant species in the wetland of Liao River estuary and a typical saline-alkaline indicator plant, most of which are distributed in coastal tidal flat, forming a rare natural landscape "pink beach" in China (Fig. 1). The main factor limiting the growth of the *Suaeda heteroptera* is the water salinity and the most suitable salinity for its growth is about 15 psu (practical salinity units): lower

than or higher than 15 psu, the *Suaeda heteroptera* will be degraded or inhibited. The salinity and water content are the main limiting factors for the growth of *Phragmites communis*, especially the salinity, and high salinity of the soil can inhibit the growth of *Phragmites communis*. Therefore, when the runoff of the river is large, the *Phragmites communis* will invade the growth area of the *Suaeda heteroptera*, resulting in community succession.



**Figure 1.** *Suaeda heteroptera* in Liao River estuary wetland

Recently, wetland ecosystems have been severely damaged and degenerated through disproportionate consumption of

wetland ecological resources, which in turn has resulted in serious declines in biodiversity and biological resources. A

5  variety of studies on hydrodynamics in estuarine wetlands have been conducted, which mainly include the following aspects:

interaction between flow and vegetation, pollutant transport in wetland and vegetation resistance experiment. The relevant

research is mainly to study the resistance coefficient of water flow when plants exist; most of them are characterized by

Manning coefficient of water resistance (Ree et al., 1958; Chow et al., 1959). In addition, some scholars have studied the

influence of plants on the structure of water flow, such as changes to flow turbulence intensity and boundary shear force

10  (Ikeda et al., 1996). Considering the height and bending degree of the willow species by water flooding, the vegetation

resistance was introduced into the Navier-Stokes equation and numerical simulation of the three-dimensional flow field of

the river and the floodplain wetland was carried out (Wilson, 2006). Taking the reed community as the research object, Shi

et al. (2001) carried out an experiment to investigate the water resistance of non-submerged reeds and the relationship

between the density and the resistance of reeds. Based on the laboratory experimental data of flow velocities for different

water depths, discharges and aquatic vegetation densities, analyses were made for the resistance coefficient of vegetation (Li

et al., 2004).

Saint-Venant equation and Nuding model were combined to simulate the steady-state and unsteady flow in the presence

of vegetation in a channel and to analyse the effects of vegetation cover, beach slope and width on the cross-section of a

river (Helmiö, 2005). A two-dimensional nonlinear hydrodynamic model (WETFLOW model) was established for wetlands

with gentle slopes and the one-dimensional and two-dimensional forms were validated by using indoor model and field pond

wetland (Feng et al., 1997). The coupled SWIFT2D surface water and SEAWAT groundwater migration model were used to

simulate the hydrological processes and salt exchange of surface water and groundwater in estuaries and adjacent coastal

wetlands (Christian et al., 2005). The two-dimensional numerical model was used to test the different flow conditions of

ZhaLong wetland, and the effect of reed wetland in the process of storage and detention was comprehensively evaluated (Gu

et al., 2006). The 2D k-ε turbulence hydrodynamic model for a curved open channel flow in curvilinear coordinates has been

set up to simulate the hydrodynamic behavior of turbulent flow in open channel partially covered with vegetation; the effect

of vegetation on flow was treated both by drag force method and equivalent resistance coefficient method (Zhang et al.,

2013). Besides, some tidal flat wetland simulations have been carried out. The Delft3D model was used to investigate the

wetland impact on tidal movement and turbulence for the semi-enclosed Breton Sound (BS) estuary in coastal Louisiana (Hu

et al., 2014). The Telemac Modelling System (TMS) was applied to the development of a hydro-environmental model of the

Severn Estuary and Bristol Channel to study microbial tracer transport processes due to mortality or interaction with the

sediments, vegetation or some other water quality constituent (Abu-Bakar et al., 2017). Stark et al. (2017) established a

depth-averaged hydrodynamic model (TELEMAC-2D) to assess the tidal hydrodynamics in marsh channels of the

Saeftinghe in the Netherlands during different stages of marsh development. Development towards a marsh system with a

channel network and a vegetated platform is strongly influenced by the pioneer vegetation. Christiansen et al. (2000)

introduced the physical processes of controlling mineral sediment deposition on a meso-tidal salt marsh surface on the

Atlantic Coast of Virginia; wetland plants patches reduce flow velocities locally and enhance sedimentation inside the

patches. Bouma et al. (2005) collected a series of hydrodynamic data from the Scheldt Estuary beaches and swamps, and

found that there was a clear linear relationship between the tidal amplitude and the maximum velocity in flats and vegetation

area, meanwhile, the flow rate was obviously lower in vegetation area. Su et al. (2013) developed a model known as

mangrove-hardwood hammock model, and simulated the evolution of vegetation succession along with changing

groundwater salinity. The results demonstrate the impact of sea level rise on coastal vegetation and groundwater salinity.

Lapetina et al. (2014) developed a 3D storm surge model with plants effect; the model is applicable to assess the feasibility

of future wetland restoration projects. Regarding salt intrusion in wetland, Andrew et al. (2017) constructed a 3-D

hydrodynamic model of San Francisco Estuary and found estuarine circulation was strongest during neap tides and unsteady

salt intrusion was strongest during spring tides. A three-dimensional hydrodynamic model (CH3D) was used to investigate

the impact of physical alteration on salinity (Sun et al., 2016).

In general, the majority of studies have focused on the effects of vegetation on fluid movement in flume experiment,

few detailed field observations or salinity simulations exist in mudflat–salt marsh ecosystems, especially in typical wetland

plant of Liao River estuary. Research on the salinity response to river discharge in wetland waters is not yet systematically

assessed.

    In this study, a 2-D hydrodynamic and salinity model is used to simulate flow patterns and salinity distribution in

wetland waters of Liao River estuary. The resistance caused by vegetation is represented by the varying Manning coefficient.

This study adopts remote sensing techniques to obtain the spatial distribution of two types aquatic plants in Pink Beach. The





numerical model is calibrated and validated against field measurement data; the variation of salinity in vegetated domain of

the Pink Beach wetland is obtained under different runoff conditions.

## 2 Numerical Models

The MIKE 21 model, one of the most widely used hydrodynamic models, was developed by Danish Hydraulic Institute (DHI)

5   and has been widely used in domestic and overseas research (Wang et al., 2013). The model is based on the cell-centered

finite volume method implemented on an unstructured flexible mesh. It includes hydrodynamic, transport, ecological

module/oil spill, particle tracking, mud transport, sand transport, and inland flooding modules (Cox, 2003).

### 2.1 Hydrodynamic module

The Hydrodynamic module is based on numerical solution of the depth-integrated incompressible flow Reynolds-averaged

10  mass conservation and momentum equations (William, 1979). The governing equations include:

Continuity conservation:

$$\frac{\partial h}{\partial t} + \frac{\partial h\bar{u}}{\partial x} + \frac{\partial h\bar{v}}{\partial y} = hS \tag{1}$$

Momentum equations:

$$\frac{\partial h\bar{u}}{\partial t} + \frac{\partial h\bar{u}^2}{\partial x} + \frac{\partial h\overline{vu}}{\partial y} = f\bar{v}h - gh\frac{\partial \eta}{\partial x} - \frac{h}{\rho_0}\frac{\partial P_a}{\partial x} - \frac{gh^2}{\rho_0}\frac{\partial \rho}{\partial x} + \frac{\tau_{sx}}{\rho_0} - \frac{\tau_{bx}}{\rho_0}$$
$$+ \frac{\partial}{\partial x}\left(hT_{xx}\right) + \frac{\partial}{\partial y}\left(hT_{xy}\right) + hu_s S \tag{2}$$

$$\frac{\partial h\bar{v}}{\partial t} + \frac{\partial h\bar{v}^2}{\partial x} + \frac{\partial h\overline{uv}}{\partial y} = -f\bar{u}h - gh\frac{\partial \eta}{\partial y} - \frac{h}{\rho_0}\frac{\partial P_a}{\partial y} - \frac{gh^2}{\rho_0}\frac{\partial \rho}{\partial y} + \frac{\tau_{sy}}{\rho_0} - \frac{\tau_{by}}{\rho_0}$$

$$+ \frac{\partial}{\partial x}\left(hT_{yx}\right) + \frac{\partial}{\partial y}\left(hT_{yy}\right) + hv_s S \tag{3}$$





where $x$ and $y$ are the Cartesian coordinates; $h = \eta + d$ is the total water depth; $t$ is time; $\eta$ is water surface elevation; $d$ is the still water depth; $\rho$ is density of water; $\rho_0$ is a ratio of water density to air density; $g$ is acceleration due to gravity; $\bar{u}$ and $\bar{v}$ are the depth-averaged velocity components in $x$ and $y$ directions; $f$ is the Coriolis parameter; $S$ is the magnitude of the discharge due to point sources; $p_a$ is the atmospheric pressure; $(u_s, v_s)$ is the velocity components in $x$ and $y$ directions for

point sources; $T_{xx}$, $T_{xy}$, $T_{yx}$ and $T_{yy}$ are the components of the effective shear stress due to turbulence and visous effects;

$(\tau_{sx}, \tau_{sy})$ and $(\tau_{bx}, \tau_{by})$ are the $x$ and $y$ components of the surface wind and bottom stresses. $\dfrac{\vec{\tau}_b}{\rho_0} = c_f \vec{u}_b \left| \vec{u}_b \right|$,

$\vec{u}_b = (u_b, v_b)$ is the depth-averaged velocity for two-dimensional calculations, $c_f = \dfrac{g}{(Mh^{1/6})^2}$, $M = 25.4 / k_s^{\frac{1}{6}}$, $M$ is

the Manning coefficient for the bed roughness in MIKE 21 model, $k_s$ is roughness height.

## 2.2 Salinity module

The fundamental salinity equation is:

$$\frac{\partial h \bar{s}}{\partial t} + \frac{\partial h \bar{u} \bar{s}}{\partial x} + \frac{\partial h \bar{v} \bar{s}}{\partial y} = h F_S + h s_s S \tag{4}$$

where $\bar{s}$ is the depth-averaged salinity under average water depth, $s_s$ is the salinity of the source, $F_S$ is the horizontal diffusion terms of the salinity.

## 3 Numerical simulation and validation

### 3.1 Description of the study domain

The Liao River is one of the largest seven rivers in China and is located in the north of the Liaodong Bay, China. This estuary is a crucial ecological economic zone, which plays an important role for the comprehensive development and

utilization of marine industry in China. The Liao River estuary includes Daliao River and Liao River (Li et al., 2017). The

Pink Beach of Liao River Delta is a marsh wetland covered with *Phragmites communis* and *Suaeda heteroptera*; it has been

listed as the largest reed wetland and the second largest marshes in the world. It provides an important habitat for a variety of

marine wildlife, especially for some endangered species, such as *Phoca largha*, *Larus saundersi* and *Grus japonensis*.

However, over the past decade, the Pink Beach wetland has been significantly degraded and *Suaeda heteroptera* community

decreased by the global warming, environmental pollution and other natural and human factors. The studies on tidal flat

wetland have shown that the growth of plant community is associated with a limited range of salinity (Zhang, et al, 2009).

However, due to the lack of quantitative salinity observations, the impact of actual salinity on vegetation growth is still

unknown for the Pink Beach wetland of Liao River.

Domain is located at the north of Liaodong Bay, extending from $40.3032\,°$ to $40.7105\,°$ North and $121.0294\,°$ to

$122.0312\,°$East (Fig. 2). An unstructured triangular mesh of Liao River estuary (Fig. 3) was generated using bathymetry data

by SMS (Surface Water Model System) software. The number of cells was 18108 with 9599 nodes in the computational

domain. From the upper reaches of the river to the central part of the domain, i.e. at the Pink Beach wetland, which is the

focus of the present study, the mesh resolution was made finer, especially close to the coastal line. Then the topographic map

was obtained through terrain interpolation (Fig. 3). The resolution of the horizontal unstructured grid was made relatively

coarser in the section of the ocean open boundary (Fig. 3).




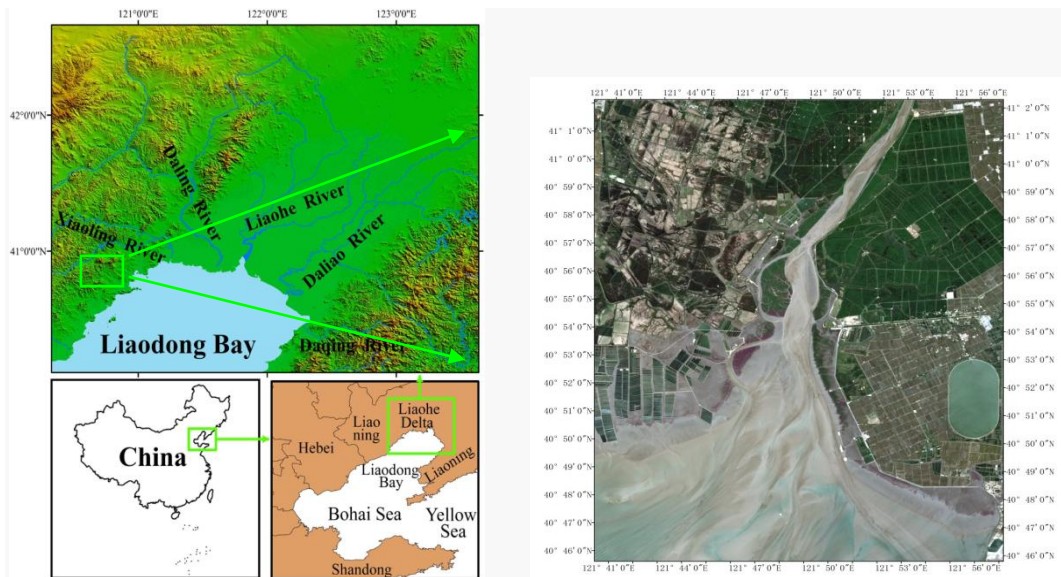

**Figure 2.** The geographical location (Wang et al., 2017) and satellite image of the Liao River Estuary

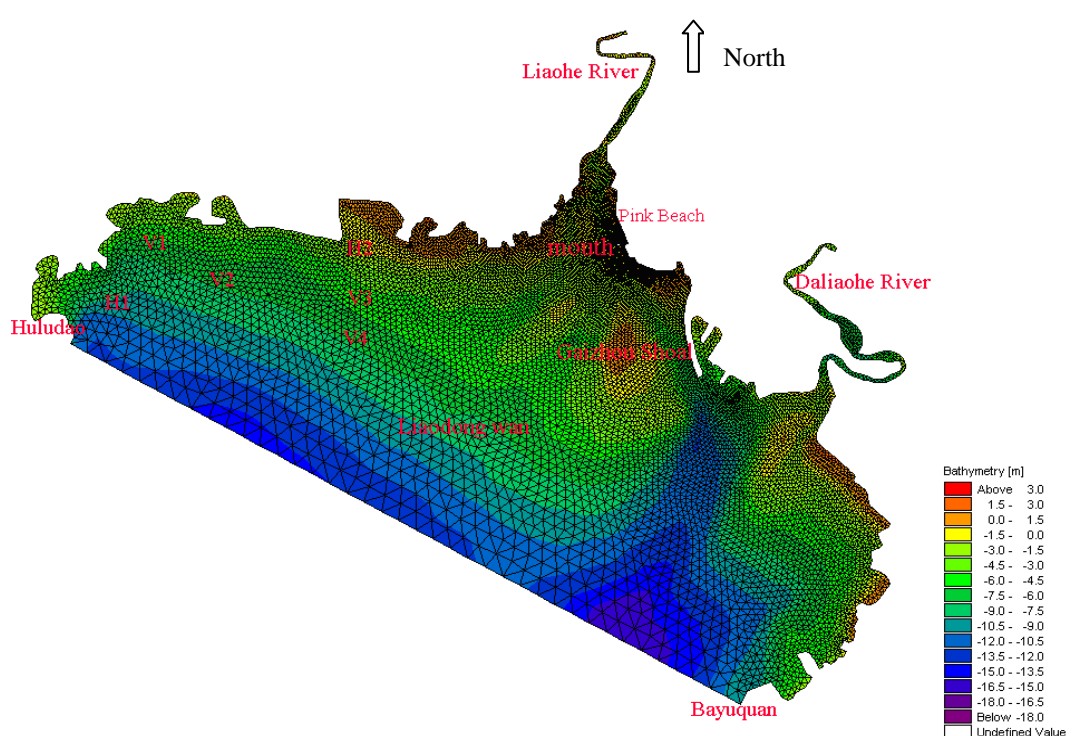

**Figure 3.** Model domain, including the mesh bathymetry and the validation points



To proceed with the numerical simulations, the equations of hydrodynamics require appropriate boundary and initial

conditions. A total of three open boundaries and the solid boundary are established. The model was forced at the open

boundary, from Huludao to Bayuquan, by a time series of tidal elevations from the TMD (Tide Model Driver) (Padman L.,

2005). Two flow boundaries in the north of area are controlled by the discharge. The solid boundary is treated as

impermeable with no slip. The salinity data of the open boundary and river discharge are set to 32.8 and 2 psu in this model.

The initial water level and salinity are 0 m and 32 psu, respectively.

**3.2 Simulation of tidal currents and salinity**

Simulation is carried out to verify the accuracy of the model. Simulations with a larger time step caused systematic

violations of the Courant Number (i.e., CN > 1), whereas smaller time steps significantly increased the computational time.

The parameters of the module are set as follows: the simulation time step was selected to be half an hour, which

automatically was interpolated to match the simulation time step. The parameter M is 80 $m^{1/3}s^{-1}$ in this study. The

hydrodynamic model was run for the period May 1, 2013, to May 31, 2014. A model spin-up period of a year was performed

to achieve stabilization from May 1, 2013, to May 1, 2014, and tidal water levels, current velocity, and salinity throughout

the water column were used as calibration parameters over the period May 21, 2014, to May 30, 2014. There was a neap tide

between May 21 and May 22, 2014, and a spring tide between May 29 and May 30, 2014. There were two tide level

monitoring stations (H1 and H2) and four tidal current monitoring stations (V1, V2, V3 and V4) (Tab. 1 and Fig. 3).

**Table 1.** The coordinate of monitoring stations

| Station | Latitude | Longitude |
|---------|----------|-----------|
| H1 | 40°47.616' | 121°04.833' |
| H2 | 40°50.151′ | 121°23.696′ |
| V1 | 40°49.226′ | 121°08.471′ |





| | | |
|---|---|---|
| V2 | 40°48.660′ | 121°15.278′ |
| V3 | 40°48.400′ | 121°24.349′ |
| V4 | 40°43.839′ | 121°23.522′ |

The water level and the tidal current in the study domain are calculated, the results of numerical simulation were compared with measured data in terms of water level and tidal currents, as shown in Fig.4, Fig.5 and Fig.6. The model matched the timing of observed tidal water levels at two locations (Fig. 4), with no detectable phase shift in water levels. The

5  simulated tidal current speeds were approximately consistent with the field data (Fig. 5 and Fig. 6). In addition, the simulated direction of tidal currents is consistent with the measured direction of tidal currents. The satisfactory validation results demonstrate that the proposed model is capable of simulating the flow in Liao River estuary.

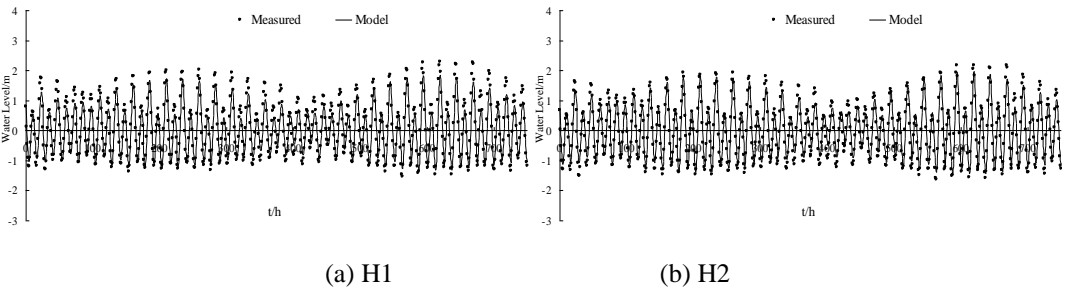

10                                         (a) H1                              (b) H2

**Figure 4.** The validation of tidal level at measured stations

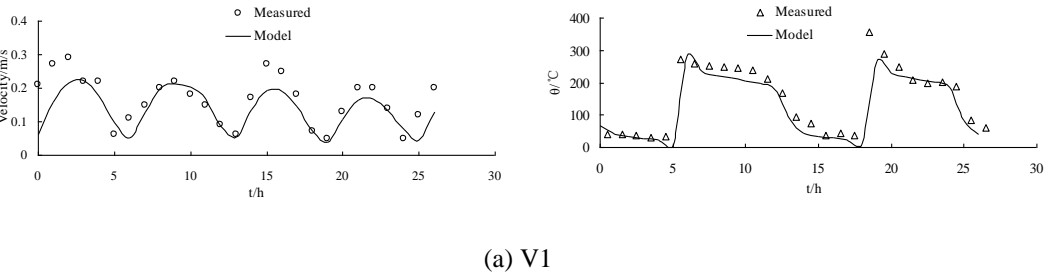

(a) V1




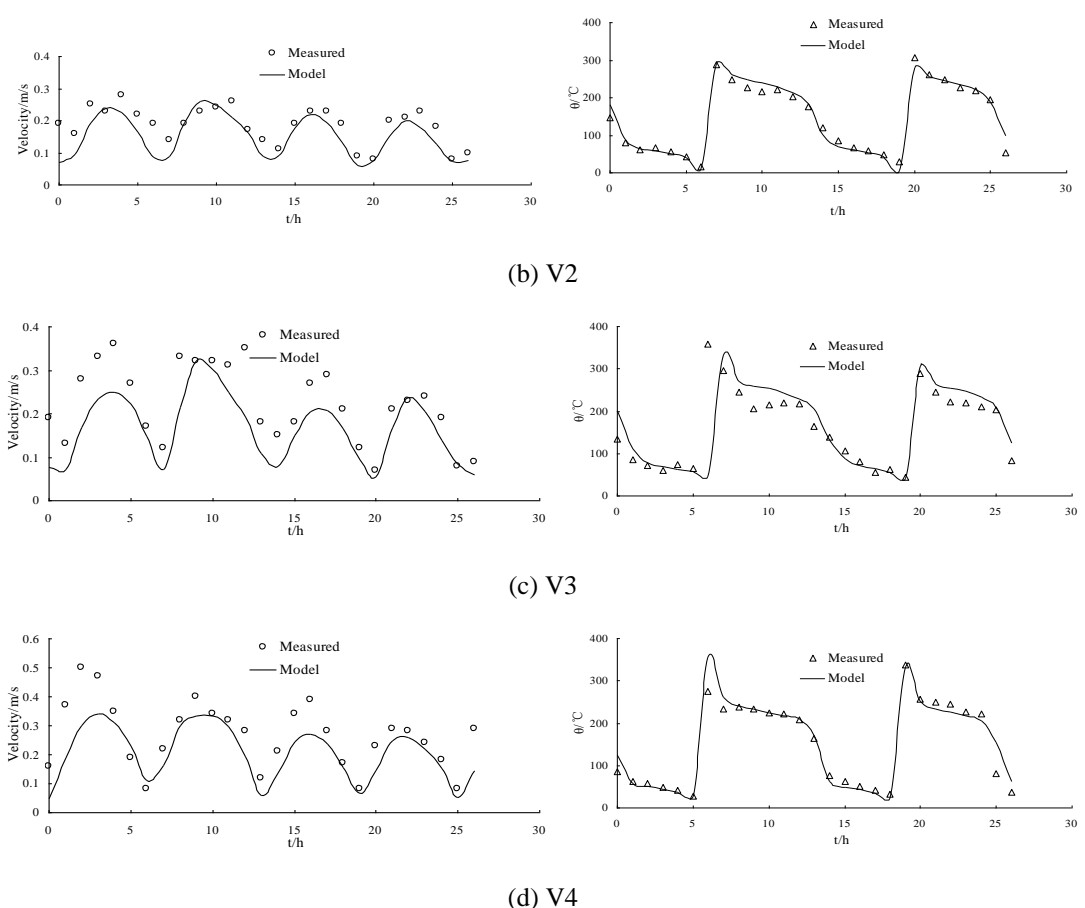

(b) V2

(c) V3

(d) V4

**Figure 5.** The validation of tidal current at measured stations during neap tide

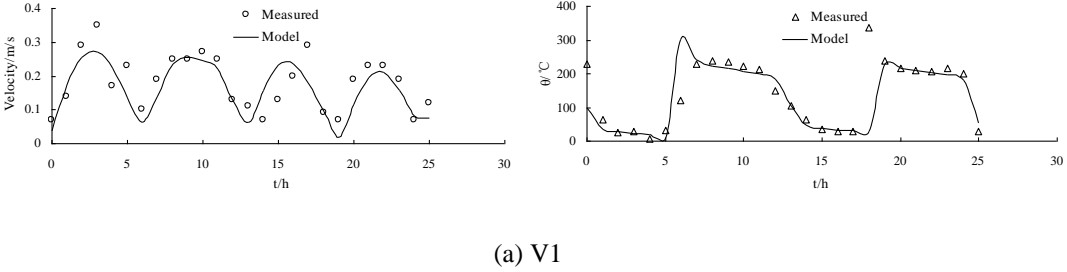

(a) V1

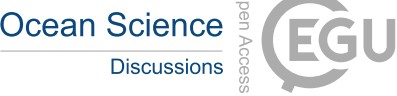

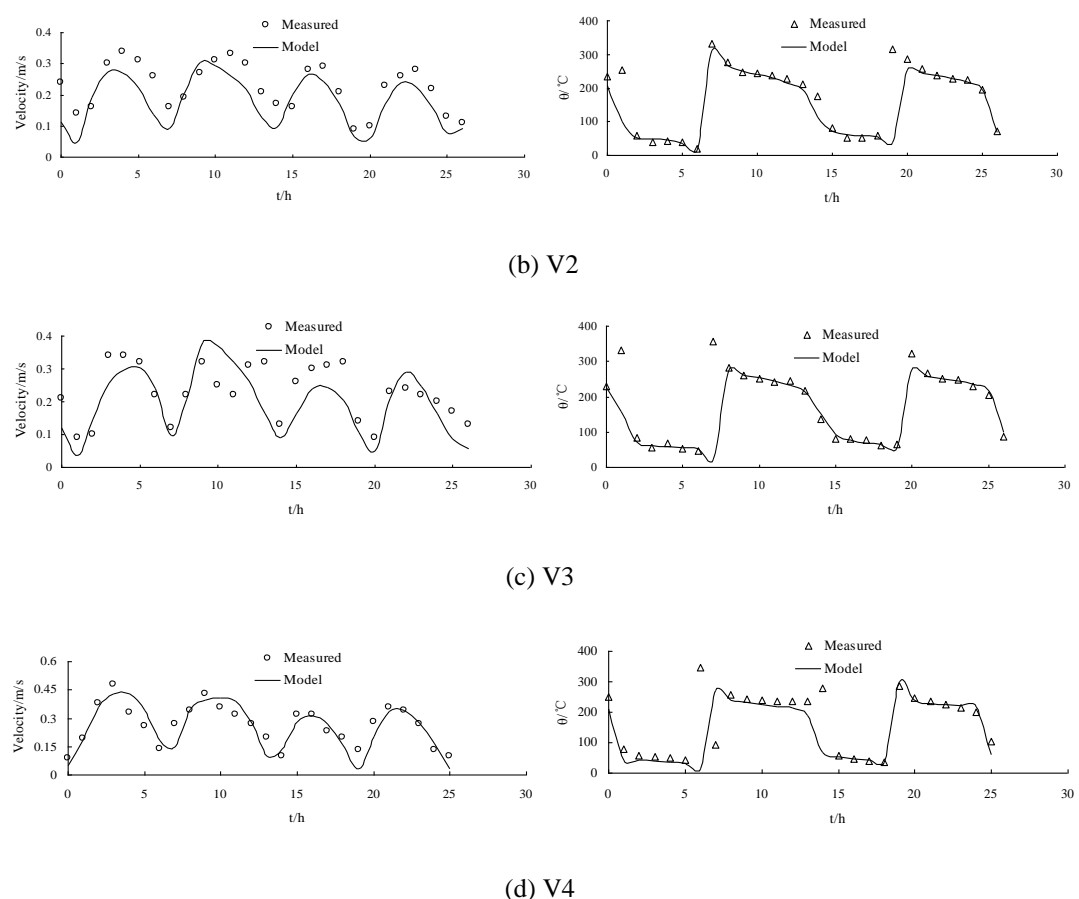

Figure 6. The validation of tidal current at measured stations during spring tide

Fig. 7 and Fig. 8 show the distribution of flow field in flood and ebb tide during the neap and spring tide. During the

flood tide, as depicted in the figure, the general flow of tidal current in Liaodong Bay is northeast. The main flow flooding

10 into Liao River divides to go around both sides of Gaizhou shoal. When the flow reaches the east and northeast of the

Gaizhou shoal, the flow turns to the northwest, forming a mainstream flow from the outside sea into Liao River. West of

Gaizhou shoal, the water flows mainly to the north and the northeast, and is affected by the delta terrain. During the spring

tide period, the Gaizhou shoal can be swamped by the tidal currents on both sides because of the high tide level. During the

ebb tide period, due to the development of many shoals, the current at the mouth of the estuary is divided into many branches.

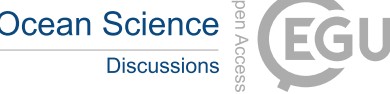



West of the Gaizhou Shoal, the current gradually turns from southwest to south. East of the Gaizhou Shoal, the tide current

flows directly to southeast and then turns to south. Part of the Gaizhou shoal is high enough to be exposed at the low tide.

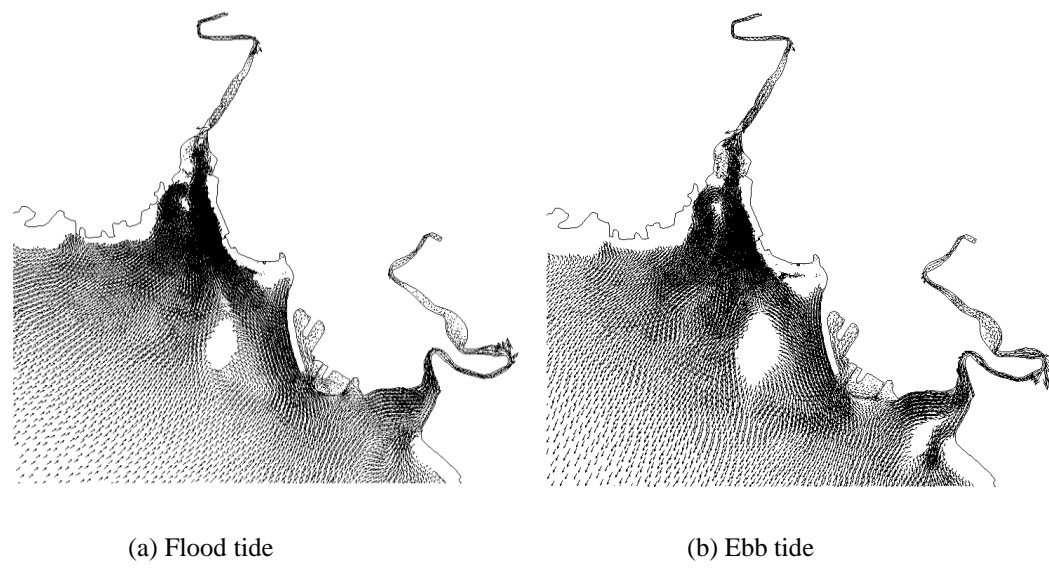

(a) Flood tide                                    (b) Ebb tide

5                                    **Figure 7.** Flow field distribution during neap tide

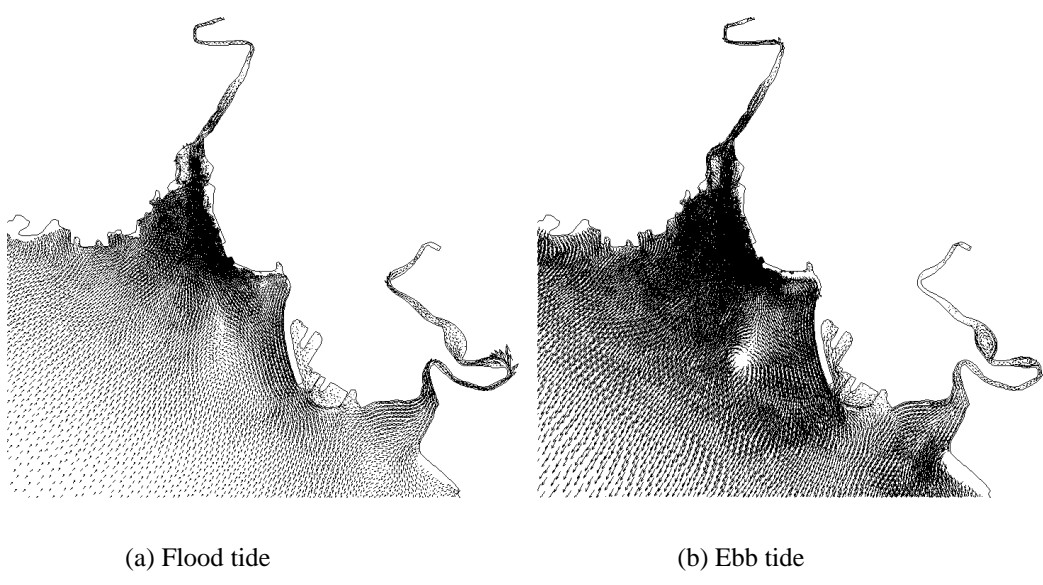

(a) Flood tide                                    (b) Ebb tide

**Figure 8.** Flow field distribution during spring tide




The results of the salinity validation during neap tide, moderate tide and spring tide are reported in Fig. 9, Fig. 10 and

Fig. 11. The model estimates correctly the salinity in the Liao River estuary.

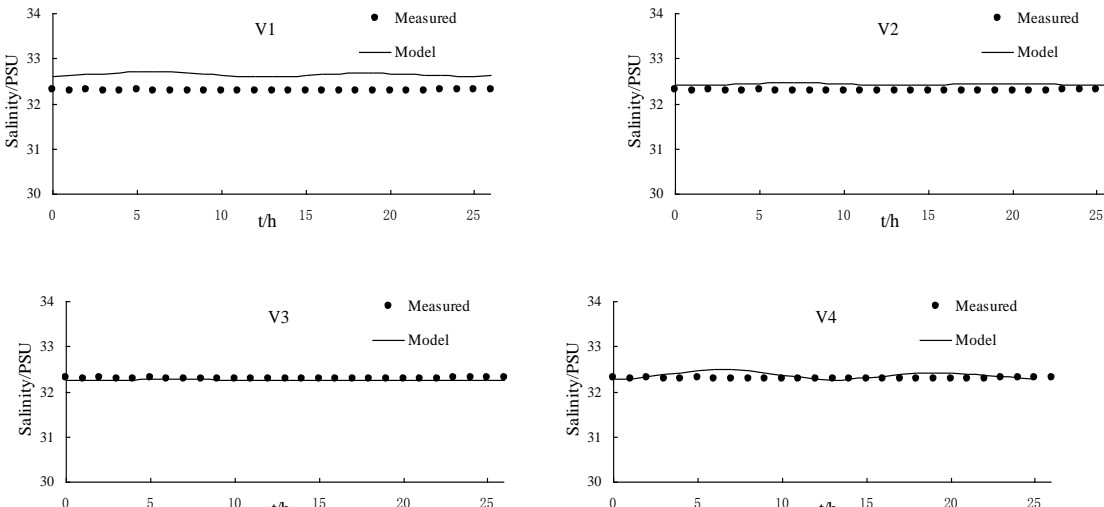

**Figure 9.** The validation of salinity at measured stations during neap tide

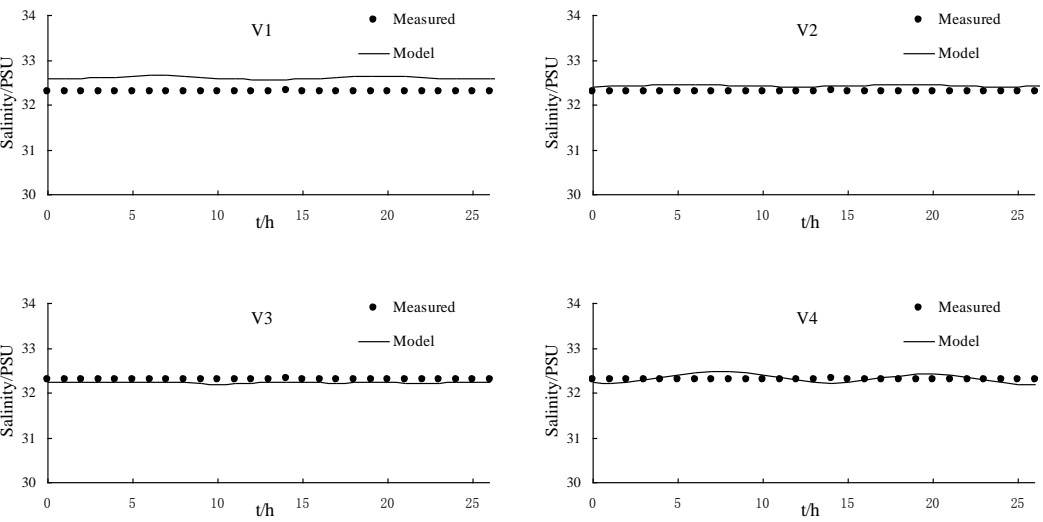



**Figure 10.** The validation of salinity at measured stations during moderate tide

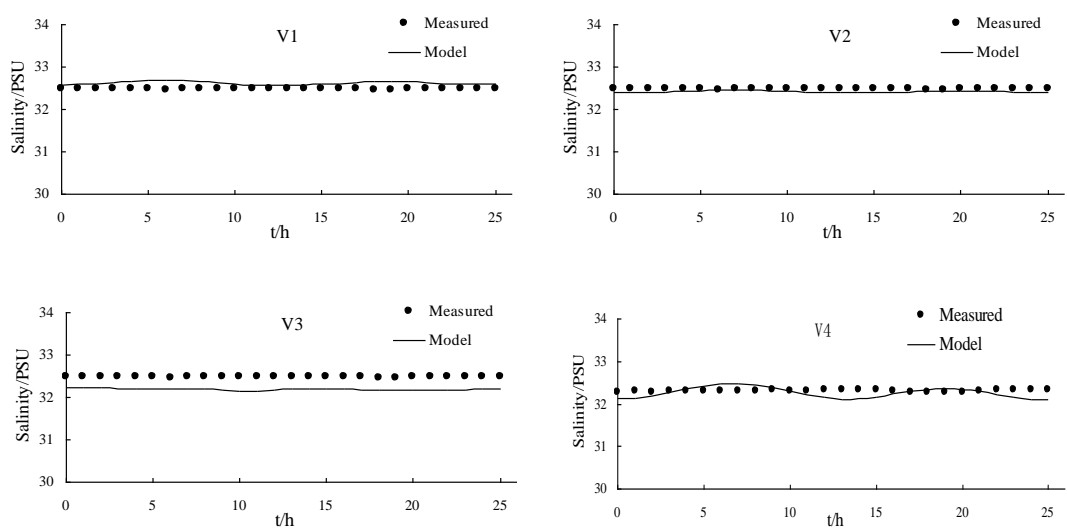

**Figure 11.** The validation of salinity at measured stations during spring tide

## 3.3 Hydrodynamics and salinity simulation in Wetland domain

### 3.3.1 Accessment of wetland information by remote sensing

Remote sensing is an effective and powerful way to monitor vegetation status, growth and biophysical parameters (Hunter et al., 2010; DeFries 2008; Ustin and Gamon 2010) and allow frequent acquisitions for multi temporal studies and reconstruction of historical time series in a cost-effective way (Coppin and Bauer 1994; Munyati 2000). The objective of the present research is to adopt remote sensing to obtain the information of vegetation in wetland of Liao River estuary.

Information on the wetland was acquired on June 3, 2017 from Landsat8 Operational Land Imager (OLI), provided by the USGS (https://glovis.usgs.gov/next/). Resolution of the images is 30m, with orbit number of 120/032. The images have undergone radiometric calibration, atmospheric correction and image cutting through ENVI 5.1 software (The Environment for Visualizing Images) before the classification. Firstly, the images were processed via the Radiometric Calibration tool, which creates the radiance images. Then, the atmospheric correction of resulting images was carried out by combining the

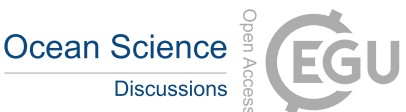

meta data (solar azimuth angle, image center latitude and longitude, data acquisition time, band gain and band deviation, etc.) through FLAASH (fast line-of-sight atmospheric analysis of spectral hypercubes) Model (Yuan, et al., 2009). The focus of this section is on the upper reaches and Pink Beach, where are vegetation-intensive areas. Therefore, the emphasis on information extraction is water body, shoal and vegetation. The NDVI ( normalized difference vegetation index ), MNDWI

( modified normalized difference water index ) and RI ( Red index ) are used to extract different objects, each index can be extracted for a class of feature information. Firstly, they were calculated based on the reflectance of each band of Landsat8 OLI sensor by using the band functions in ENVI software. Then, different thresholds were set to classify different features. Finally, the decision tree classification (Fig. 12) in ENVI classification tool was executed to realize the extraction of the water body, shoal and vegetation. Vegetation is divided into two categories: *Phragmites communis* and *Suaeda heteroptera*

(Fig. 13). The stations of G1, G2, P1, P3 and P5 are presented in Fig.13.

$$\mathrm{MNDWI} = (b3 - b6)/(b3 + b6) \tag{5}$$

$$\mathrm{NDVI} = (b5 - b4)/(b5 + b4) \tag{6}$$

$$RI = (b4 - b3)/(b4 + b3) \tag{7}$$

b3 is the green band reflectance of Landsat8 OLI sensor, b4 is the red band reflectance, b5 is the near infra-red band

reflectance, b6 is the middle infra-red band reflectance.



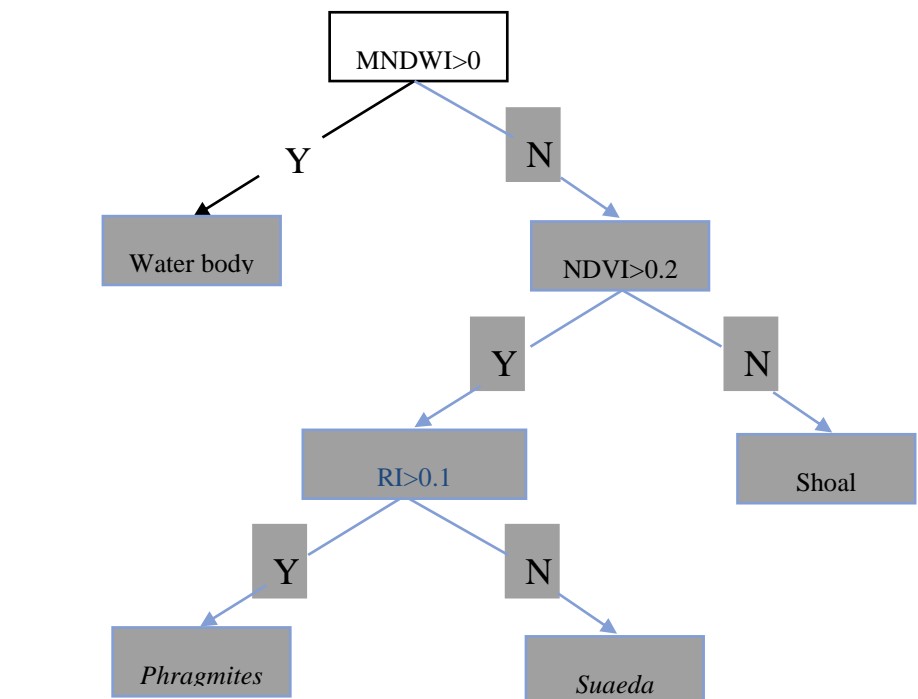

**Figure 12.** Vegetated classification based on decision tree

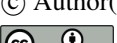


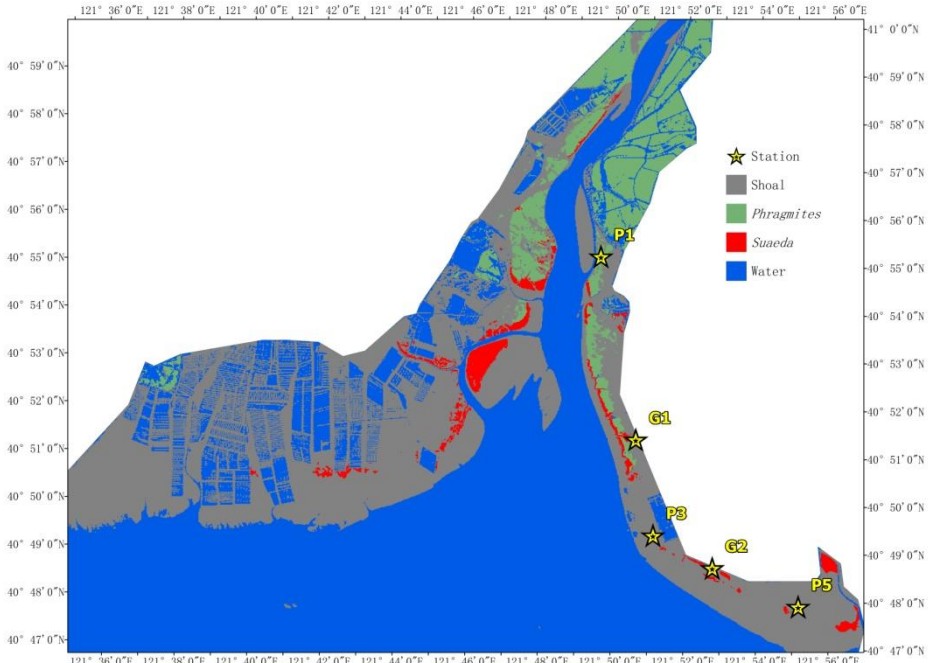

**Figure 13.** The distribution of aquatic plant and selected stations in Pink Beach

The effect of vegetation on hydrodynamics in areas of wetland in estuary was represented by a varying Manning's coefficient in the bottom friction term. The Manning's coefficient for the vegetation resistance depends on the flow depths,

5     number density and the diameter of the vegetation elements (Zhang et al., 2013). The Manning's coefficient $n_v$ considering vegetation effect is given by

$$n_v = \sqrt{\left(\frac{1}{M}\right)^2 + \frac{C_D m D \min(h, h_v) h^{1/3}}{2g}}$$

(8)

where $m$ is the number density (the number of vegetation elements per unit horizontal area), $m = \dfrac{1}{d_v}$, where $d_v$ is the

average distance between two adjacent vegetation elements, $C_D$ is the drag coefficient, $D$ is the averaged diameter of the

10     vegetation element, $h_v$ is vegetation height.

**3.3.2 Hydrodynamics and salinity simulation in Wetland domain**

The model simulation was conducted to evaluate the estuarine hydrodynamics and salinity transport in the presence of

vegetation at the Pink Beach wetland, incorporating realistic vegetation in the model grid. The dominant vegetation at the

sites (Fig. 13) is *Phragmites communis* and *Suaeda heteroptera*. The averaged diameters of plants are 0.6 cm and 0.2 cm,

respectively; the plant stems are set to 1.5 m and 0.15 m high, respectively. The drag coefficients in the model ($C_D$) are set to

1.0 and 0.3, respectively. The density is set to 65 per square meter for *Phragmites communis* and 200 stems per square meter

for *Suaeda heteroptera,* respectively (He et al., 2008). The simulated and measured changes in water depth and salinity

concentration at two stations (G1 and G2) are demonstrated in Fig. 14 and Fig. 15. Water depth in the Pink Beach region is

correctly modelled, but at G1 upstream of the vegetation zone, this model is in error compared to the observed results. The

model predicts salinity concentration reasonably well compared with the measured data at the Pink Beach. The maximum

water depth of G1 is 0.834 m during the spring tide on June 29, the immersion time is 282 minutes; the maximum salinity is

31.02 psu. At G2, the maximum water depth is 0.682 m during the spring tide on July 26 to 27, the immersion time is 184

minutes; the maximum salinity is 34.76 psu. Velocity comparison between the modeled and measured value at G1 station is

shown in Fig. 16; this model is relatively accurate. The velocity of G1 displays the shape of double humps, its peak depth-

averaged velocity can reach 0.15 ms⁻¹. The results of flow structure at Pink Beach in presence and absence of vegetation

highlight the relationship between vegetation and currents (Fig.17). From the numerical experiments, it can be seen that the

presence of vegetation increases the resistance of the estuary bed and can effectively reduce the flow velocity. This is

because when water flows through the vegetation, momentum and energy are lost, the drag exerted by vegetation results in

decreased flow speed.

According to the measured and simulated salinity data of Liao River estuary during the spring tide on July 26 to 27,

2017, five stations in the Pink Beach wetland were selected from upstream to downstream to analyze the longitudinal



distribution of salinity in the tidal cycle under the same runoff conditions as shown in Fig.13. The simulated salinity data for

several stations along the Liao River from the entrance are given in Fig. 18, the salinity concentration at P1 upstream is far

lower than that of P5 downstream, the salinity concentration increases from upstream to downstream in Liao River. As

dilution by fresh water from upstream increases, salinity decreases. During the ebb tide, the negative tide levels occur with

5    the dry domain at locations of G1, G2, P3 and P5, so the salinity concentrations show gaps for this period. The influence of

runoff variation on the salinity distribution of G1 and G2 in Pink Beach wetland was analyzed in different runoff conditions

0 $m^3s^{-1}$, 30 $m^3s^{-1}$, 101 $m^3s^{-1}$, 285 $m^3s^{-1}$ and 450 $m^3s^{-1}$, respectively. As depicted in Fig. 19, the salinity concentration of Pink

Beach in 0 $m^3s^{-1}$ is significantly higher than that of other cases. The salinity of G1 and G2 reaches about 32.78 psu with the

same value in the case of 0 $m^3s^{-1}$; the salinity of G1 and G2 decreases to about 7.26 and 16.7 psu with 101 $m^3s^{-1}$; when the

10   runoff is 450 $m^3s^{-1}$, the salinity concentration of G1 is only 2 psu, and the salinity concentration of G2 is about 5.66 psu. The

impact of discharge on salinity distribution in the Liao River estuary is fairly remarkable. During the wet season, due to the

higher water discharge, concentrations at G1 and G2 are relatively lower. During the dry season, the flux of fresh water

discharged into Pink Beach declines substantially, which leads to enhanced saltwater intrusion. That is to say that the salinity

concentration decreases with the increase of inflow runoff.

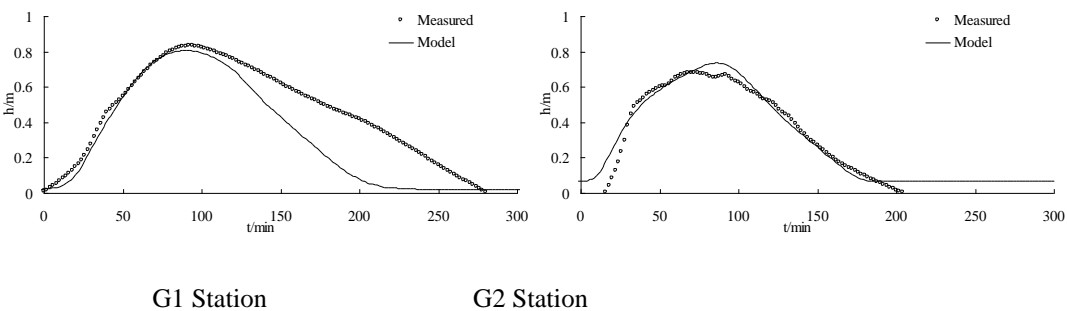

G1 Station                        G2 Station

**Figure 14.** Comparison of the simulated and measured water depth at location G1 and G2

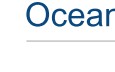
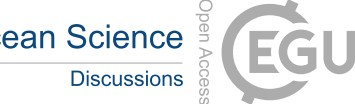

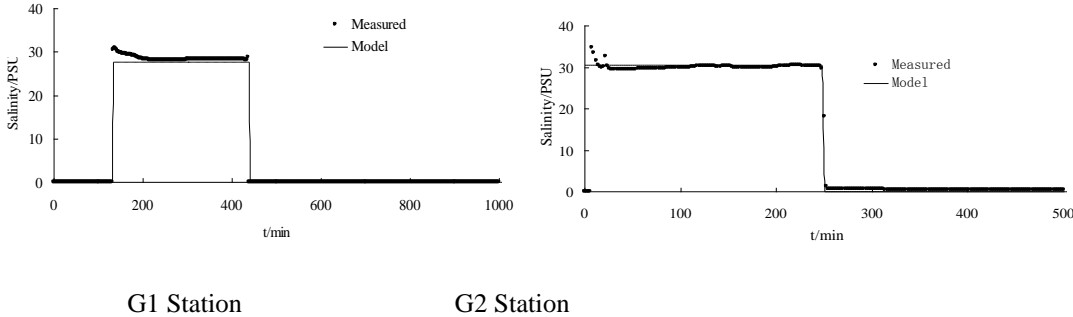

G1 Station                    G2 Station

**Figure 15.** Comparison of the simulated and measured salinity concentration at location G1 and G2

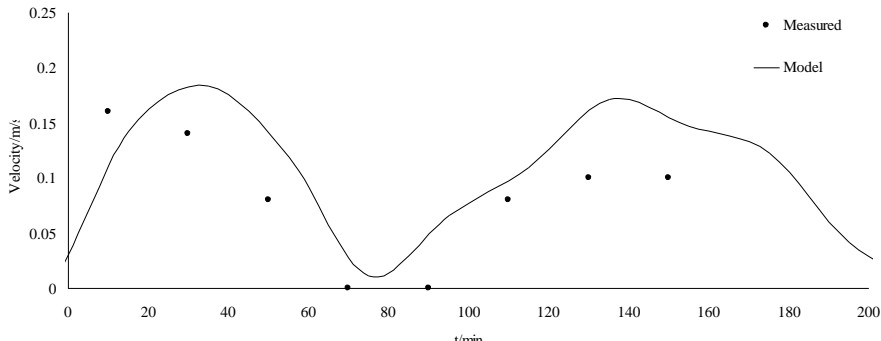

**Figure16.** Comparisons of the measured and simulated velocities at location G1





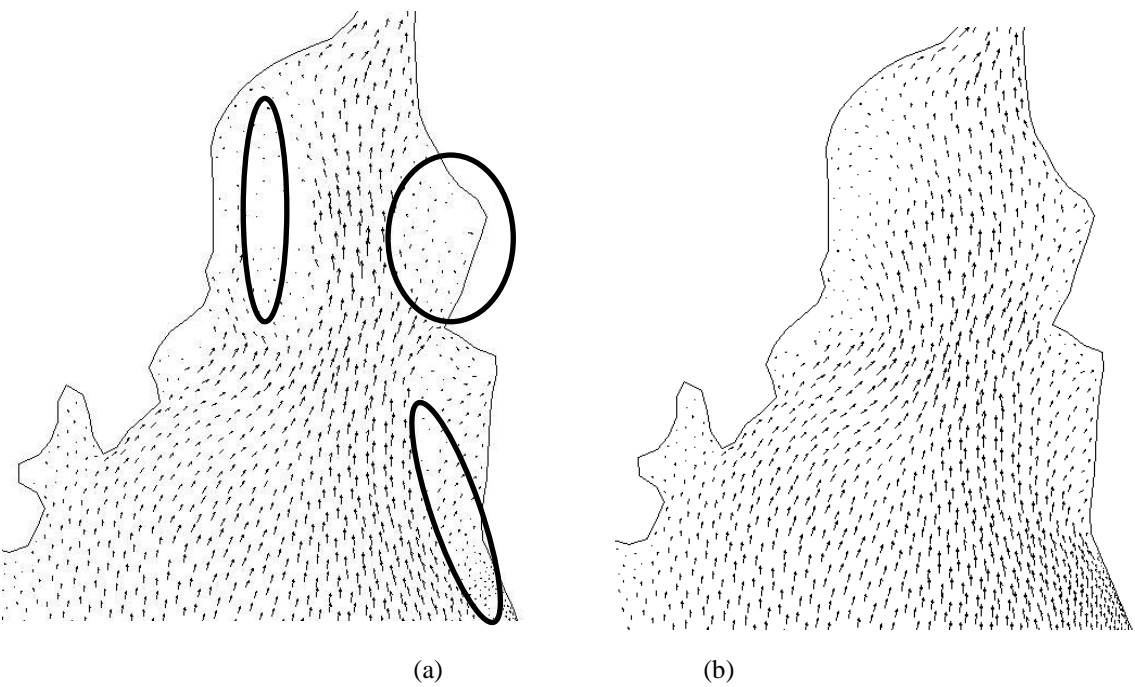

       (a)           (b)

**Figure 17.** Flow structure of Pink Beach in vegetated and non-vegetated area, the black ellipses represent vegetation areas

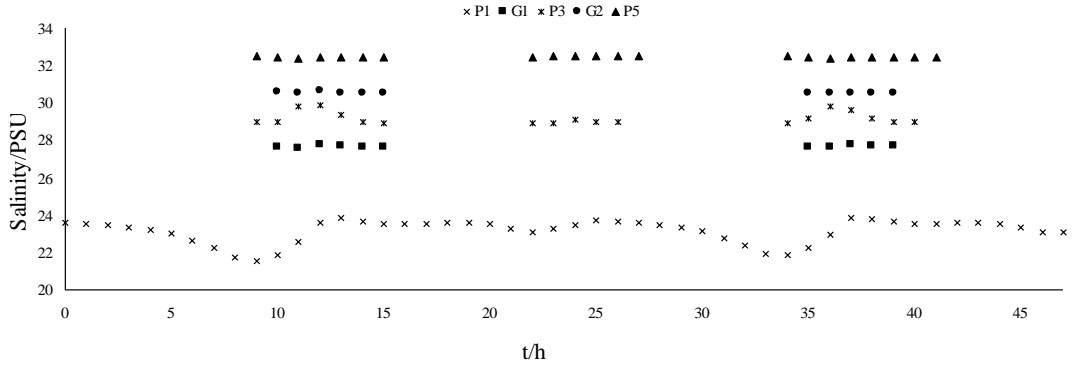

**Figure 18.** The simulated salinity concentration at five locations




**Figure 19.** Simulated salinity with different runoff

## 4 Discussion

5   In this study, the MIKE 21 model is used to simulate the hydrodynamic characteristics and salinity transport process in Pink

Beach wetlands of Liao River estuary. The model couples the hydrodynamic and salinity modules with aquatic plant effect.

The spatial discretization of the primitive equations is performed using a cell-centered finite volume method in the horizontal

plane with an unstructured grid of triangular elements. Landsat images are applied to differentiate the wetland vegetation

types in the Liao River estuary. Based on the obvious spectral distinction of vegetations, a decision tree containing a number

of decision rules is designed to classify different types of vegetation cover; the Liao River estuary is classified into water body, shoal, and major wetland vegetation types, e.g., *Phragmites communis* and *Suaeda heteroptera*.

The model is tested by simulating the water level, tidal current and salinity concentration in Liao River estuary, and the results are consistent with the measured data. The tidal flats are periodically exposed above the surface of the water in

Liao River estuary. Numerical predictions indicate that vegetation imposes significant influence on flow dynamics. The existence of vegetation is associated with lower flow velocities, the vegetation can modify the flow structure owing to energy dissipation induced by vegetation. By analyzing the longitudinal variation of salinity in Pink Beach wetland, we found that salinity gradually increased from upstream to downstream. The effect of runoff on salinity distributions in the Pink Beach is fairly distinct. When the river discharge is low, the mixture of the upstream freshwater is weak and salinity is

greater. It is important to understand the wetland dynamics and salinity transport process, and this research contributes to an improved understanding of suitable circumstances for the vegetation growth in Pink Beach. More generally, this study can provide an important scientific basis for wetland conservation and restoration.

**Acknowledgements.** This work was supported by the National Nature Science Foundation of China (51579030), the Wetland Degradation and Ecological Restoration Program of Panjin Pink Beach (PHL-XZ-2017013-002), the Fund of

15 Liaoning Marine Fishery Department (201725), the Open Fund of the State Key Laboratory of Hydraulics and Mountain River Engineering (SKHL1517).

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
