# Peer review of "Numerical study of hydrodynamic and salinity transport process in Pink Beach wetlands of Liao River Estuary, China"

_Ocean Science, 2017_

## Referee Comment (RC1) · Anonymous Referee #1 · 23 Feb 2018

Manuscript Number: OS-D-2017-102 Title: Numerical study of hydrodynamic and salinity transport process in Pink Beach wetlands of Liao River estuary, China This study uses MIKE 21 hydrodynamic and salinity model to simulate the hydrodynamic characteristics and salinity transport process in Pink Beach wetlands of Liao River estuary, the effect of wetland plant on tidal flow is considered there. The remote sensing techniques is adopted to acquire vegetation distribution is based on Landsat TM satellites in this study. This study of manuscript is very interesting, it is well written on the whole and provides researchers with useful information and knowledge about wetland hydrodynamics. There are very few typographical and figures errors in this manuscript. In general, I believe that with major revisions this paper would be of interest to readers

of OS. I recommend this manuscript to publish in this Journal with a minor revision, the revisions are listed below: 1. Please indicate the location of Gaizhou Shoal in figure 3. 2. Page 8, what's the meaning of us and vs? 3. What does a blank area represent in Figure 7? Please state. 4. In Figure 13, the five stations of G1, G2, P1, P3 and P5 are not clear, please edit it. 5. As can be seen from Figure 14, there is an asymmetric feature of the flood and ebb tide in the vegetation area. Please give a discussion. 6. In figure 19, it should be m3/s. 7. In addition, authors should carry out a sensitivity analysis for comparison of the velocities over plant with different plant density and different Cd in wetland water. 8. Page 4, line 14: 'Based on the laboratory experimental data of flow velocities. . . . . .' should be 'Based on the experimental data of flow velocities. . . . . .'

---

## Referee Comment (RC2) · Anonymous Referee #2 · 26 Mar 2018

"Numerical study of hydrodynamic and salinity transport process in Pink Beach wetlands of Liao River Estuary, China"

This paper applies the Mike unstructured mesh model to the Liao River. The aim of the model investigation is to understand how the presence of vegetation impacts the flow, and salt intrusion in the estuary. Landsat imaginary is used to derived the vegetation cover input to the model, then Mike is used to simulate hydrodynamics and salinity.

Main suggestion Salinity is supposed to be the key result, so please map it, and discuss it more thoroughly. I would like to see more analysis of the type shown in figure 17. The impact of vegetation on the flow, but also on the salinity. How does the map of

salinity look in these 2 simulations? How do the patterns change with different seasons, comparing wet/dry discharge? How different are the maps at high/low tide?

You mention P25, line 9 - that there is more salinity intrusion when discharge is low - but you have not really evidenced this in your paper. The modelling work is nice, and the impacts of vegetation are an interesting subject for study. However, overall the paper is pretty simple, and needs more analysis of the results to be more well-rounded.

Minor comments P8 model domain, what is the max/min element size? P11, Figure4. It looks like the tidal range is a little underpredicted in the model.

Please add some simple statistics to the assessment of water level, e.g. correlation, mean-square error. What is being shown in the right hand panels of figure 5 and 6? Is is the current direction? If so, is it just current speed on the left, not velocity? Legends need expanding to make this clear.

Figure 8 is hard to read. Maybe better to plot onshore/offshore current speeds? or reduce the number of vector arrows. Also there is no key, so it is impossible to interpret the absolute magnitude in figures 7 and 8.

Figures 9 and 10 are unnecessary, you could just make a 1 line mention of the mean and standard deviation to show that the salinity is well captured.

p19 having M for manning coefficient (P7). Then defining nv as the vegetation manning roughness is a little confusing, with m then as the number of vegetation elements. It is all consistent and correct, but different naming might make it more straightforward. P20, line 14 'model is relatively accurate' - not quantitative, some statistics please.

The experiment is using different river run-off rates, defined on P21, line 6-7. To me, this is the most interesting part, and could be expanded. For example, I would suggest adding a map, showing the position of salinity contours, with different discharge rates.

Figure 17: Again, this is a really interesting result, the impact of vegetation on the flow: I think this is the key figure, and could do with more explanation in the legend. It also

needs a key again, showing the speed indicated with each vector arrow. Could also highlight where the vegetation is in the 2 different configurations.

Figure 18: I don't learn anything from this figure, other than the salinity remains relatively constant at all sites. If you keep this figure in, you need a map to show where these sites are, instead I would suggest showing a contour map of salinity across the whole domain.

Figure 19 - again cut this, and just explain in the text.

---

## Author Comment (AC1) · 26 Mar 2018

Response to evaluations of Reviewer #1: We agree with your major comments and suggestions. The replies are follows: 1. The location of Gaizhou Shoal in figure will be repainted in revised manuscript. 2. Page 8, us and vs is the velocity components in x and y directions for point sources in this manuscript. 3. In Figure 7, a blank area represents the part of the Gaizhou shoal is high enough to be exposed at the low tide, it will be explained in revised manuscript. 4. In Figure 13, the five stations of G1, G2, P1, P3 and P5 have been edited and was cleared. 5. The Liao River estuary has a complex terrain with a large area of tidal flats and shoals, which is one of the

causes of the asymmetry of flood and ebb tide. In addition, there are a large number of Phragmites communis and Suaeda heteroptera in Pink Beach, the resistance induced by vegetation enhance the tide asymmetry. 6. In figure 19, the error will be modified in revised manuscript. 7. The mentioned part will be added to the revised manuscript 8. 'Based on the laboratory experimental data of flow velocities. . . . . .' will be revised 'Based on the experimental data of flow velocities. . . . . .'

―――――――――――――――――――――――

---

## Referee Comment (RC3) · Anonymous Referee #1 · 1 Apr 2018

I thank the authors for their reply to my concerns. The point-by-point response and rebuttal were convincing. So this manuscript may be accepted after further minor revision.

---

## Referee Comment (RC4) · Anonymous Referee #1 · 1 Apr 2018

I thank the authors for their reply to my concerns. The point-by-point response and rebuttal were convincing. So this manuscript may be accepted after further minor revision.

---

## Author Comment (AC2) · 2 Apr 2018

The comment was uploaded in the form of a supplement:
https://www.ocean-sci-discuss.net/os-2017-102/os-2017-102-AC2-supplement.zip

---

## Author Comment (AC3) · 2 Apr 2018

Response to evaluations of Reviewer #2: We agree with your major comments and suggestions. The replies are follows:

1. We have simulated the impact of vegetation on water depth and salinity in this coastal wetland, the results show that the effects of wetland vegetation on water depth and salinity of wetland domain is not obvious, it can be seen in Fig. 1 and Fig.2. The Liao River estuary has a complex terrain with a large area of tidal flats and shoals, where the vegetation grow up. During the ebb tide, vegetation area is not submerged by tide. Only in spring tide, the vegetation can be merged and its salinity is influenced

by the water from open sea. From the Fig. 1and Fig.2, one can see that water depth in vegetation area does not change significantly compared to non-vegetated, and the variation of salinity is not obvious.

As the reviewer mentioned, whether it is in the wet season or dry season, the discharge has a great influence on salinity. The authors have finished the simulations about salinity concentration in Liao River estuary, Fig. 3. and Fig. 4 show that the patterns of salinity change with wet season and dry season. But due to limit length of this manuscript, it has not been provided in this study. If necessary, we will add this part to the revised version.

2. As shown in Fig. 3. and Fig. 4, there is more salinity intrusion with low discharge, we will make a comprehensive analysis that the effect of vegetation on flow and salinity in revised version.

3. We refined the model grid from the upper reaches of the river to the central part of the domain, i.e. at the Pink Beach wetland, which is the focus of the present study, with finer grid resolutions, as small as 98 m. Model grid size increases gradually away from the flats and estuarine deltas, and max cell reaches about 2460 m near the open boundary. Page 11, in figure 3, the tidal range is a little under predicted in the model, it may be related to the inaccurate tidal level extracted by the open boundary and the accuracy of the measured value. These reasons will be added to the revised manuscript.

4. The assessment of water level, e.g. correlation, mean-square error will be given in revised manuscript. We have explained that the figures 5 and 6 represent the currents, the left is current speed and the right is current direction. We will expand to make it clear in revised manuscript.

5. In figure 8, the left-hand panel shows the distribution of flow field in flood tide during the spring tide. In figures 7 and 8, the number of vector arrows will be reduced to make them clear and reference speed will be added to revised manuscript.
6. Considering reviewer suggestion, the figures 9 and 10 may be removed in revised manuscript.

7. Page 19, in MIKE 21 hydrodynamic and salinity model, M is Manning's coefficient for bed roughness, nv is Manning's coefficient for the vegetation resistance, M and nv are reciprocal. In this paper, m represents the vegetation density, we can name it in another name in case of confusing with M. Page 20, line 14, the model has been verified in the previous article. We will increase more statistics to test the accuracy of the model in Pink Beach wetland in revised manuscript.

8. We have mapped that the position of salinity contours with different discharge rates (Fig. 5, Fig. 6 and Fig. 7), but didn't put it into this manuscript due to the limited writing.

9. According to the reviewer's suggestion, we will make more discussions about the impact of vegetation on the flow, and add an arrow to indicate the speed. In figure 17 (a), the black ellipses represent vegetation areas, the other is non-vegetated area.

10. The figure 18 showed that five stations in the Pink Beach wetland were selected from upstream to downstream to analyze the longitudinal distribution of salinity in the tidal cycle under the same runoff conditions, the location of P1, G1, P3, G2 and P5 is shown in figure 13.

11. Figure 19 estimates the salinity variation of G1 and G2 under different river runoffs in Pink Beach. Obviously, the effect of runoff on salinity distributions in the Pink Beach is fairly distinct. When the river discharge is low, the mixture of the upstream freshwater is weak and salinity is greater.
* * *
[Figure]

**Fig. 1.**

[Figure]

**Fig. 2.**

[Figure]

**Fig. 3.**

(a) Dry season

Salinity [PSU]
Above 32.5
30.0 - 32.5
27.5 - 30.0
25.0 - 27.5
22.5 - 25.0
20.0 - 22.5
17.5 - 20.0
15.0 - 17.5
12.5 - 15.0
10.0 - 12.5
7.5 - 10.0
5.0 - 7.5
2.5 - 5.0
0.0 - 2.5
-2.5 - 0.0
Below -2.5
Undefined Value

[Figure]

Salinity [PSU]
Above 32.5
30.0 - 32.5
27.5 - 30.0
25.0 - 27.5
22.5 - 25.0
20.0 - 22.5
17.5 - 20.0
15.0 - 17.5
12.5 - 15.0
10.0 - 12.5
7.5 - 10.0
5.0 - 7.5
2.5 - 5.0
0.0 - 2.5
-2.5 - 0.0
Below -2.5
Undefined Value

(b) Wet season

**Fig. 4.**

[Figure]

Salinity [PSU]
Above 32.5
30.0 - 32.5
27.5 - 30.0
25.0 - 27.5
22.5 - 25.0
20.0 - 22.5
17.5 - 20.0
15.0 - 17.5
12.5 - 15.0
10.0 - 12.5
7.5 - 10.0
5.0 - 7.5
2.5 - 5.0
0.0 - 2.5
-2.5 - 0.0
Below -2.5
Undefined Value

(a) $101 m^3/s$

**Fig. 5.**

Salinity [PSU]
Above 32.5
30.0 - 32.5
27.5 - 30.0
25.0 - 27.5
22.5 - 25.0
20.0 - 22.5
17.5 - 20.0
15.0 - 17.5
12.5 - 15.0
10.0 - 12.5
7.5 - 10.0
5.0 - 7.5
2.5 - 5.0
0.0 - 2.5
-2.5 - 0.0
Below -2.5
Undefined Value

(b) 285m³/s

**Fig. 6.**

Salinity [PSU]
- Above 32.5
- 30.0 - 32.5
- 27.5 - 30.0
- 25.0 - 27.5
- 22.5 - 25.0
- 20.0 - 22.5
- 17.5 - 20.0
- 15.0 - 17.5
- 12.5 - 15.0
- 10.0 - 12.5
- 7.5 - 10.0
- 5.0 - 7.5
- 2.5 - 5.0
- 0.0 - 2.5
- -2.5 - 0.0
- Below -2.5
- Undefined Value

(c) 450m³/s

**Fig. 7.**

---

## Author Response (AR1)

Dear editors and reviewers:

We thank the valuable comments provided by two reviewers, and try our best to revise the manuscript. The overall responds to the reviewer's comments are listed as following:

To reviewer #1:

1. We have corrected the minor mistakes that you proposed.

2. In Figure 7, a blank area represents the part of the Gaizhou shoal is high enough to be exposed at the low tide, it has been explained in revised manuscript.

3. The tidal asymmetry of the wetland vegetation area has also been explained in the revised manuscript

4. We have carried out a sensitivity analysis for comparison of the velocities over plant with different plant density in wetland water.

To reviewer #2:

1. The max/min element size has been added to the manuscript.

2. In revised manuscript, we have simulated and analyzed the impact of vegetation on water depth and salinity in this coastal wetland.

3. As reviewer mentioned, whether it is in the wet season or dry season, the discharge has a great influence on salinity. The authors have finished the simulations about salinity concentration in Liao River estuary with different discharge. Authors have illustrated the contours of salinity and presented the reason for the impact of runoff on salinity in the paper.

4. Considering reviewer suggestion, the figures 9 and 10 have been removed in the paper.

5. In figures 5 and 6, the current speed and current direction have been clarified.

6. In figures 7 and 8, the number of vector arrows has been reduced and reference speed has been added to this paper.

7. Some discussion has been done for thoughtful explanation for vegetation effect on flow and salinity of water.

Thank you for your evaluation of the manuscript again.
The authors would like to revise this manuscript if reviewers have any other questions.

Best regards
On behalf of all authors

[revised manuscript text omitted]

$$\frac{\partial h\overline{v}}{\partial t} + \frac{\partial h\overline{v}^2}{\partial x} + \frac{\partial h\overline{uv}}{\partial y} = -f\overline{u}h - gh\frac{\partial \eta}{\partial y} - \frac{h}{\rho_0}\frac{\partial P_a}{\partial y} - \frac{gh^2}{\rho_0}\frac{\partial \rho}{\partial y} + \frac{\tau_{sy}}{\rho_0} - \frac{\tau_{by}}{\rho_0}$$

$$+ \frac{\partial}{\partial x}\left(hT_{yx}\right) + \frac{\partial}{\partial y}\left(hT_{yy}\right) + hv_sS \tag{3}$$

)

where $x$ and $y$ are the Cartesian coordinates; $h = \eta + d$ is the total water depth; $t$ is time; $\eta$ is water surface elevation; $d$ is the still water depth; $\rho$ is density of water; $\rho_0$ is a ratio of water density to air density; $g$ is acceleration due to gravity; $\overline{u}$ and

5    $\overline{v}$ are the depth-averaged velocity components in $x$ and $y$ directions; $f$ is the Coriolis parameter; $S$ is the magnitude of the discharge due to point sources; $p_a$ is the atmospheric pressure; $(u_s, v_s)$ is the velocity components in $x$ and $y$ directions for point sources; $T_{xx}$, $T_{xy}$, $T_{yx}$ and $T_{yy}$ are the components of the effective shear stress due to turbulence and visous effects; $(\tau_{sx}, \tau_{sy})$ and $(\tau_{bx}, \tau_{by})$ are the $x$ and $y$ components of the surface wind and bottom stresses. $\dfrac{\vec{\tau}_b}{\rho_0} = c_f \vec{u}_b \left|\vec{u}_b\right|$, $\vec{u}_b = (u_b, v_b)$ is the depth-averaged velocity for two-dimensional calculations, $c_f = \dfrac{g}{(Mh^{1/6})^2}$, $M = 25.4/k_s^{\frac{1}{6}}$, $M$ is the

10    Manning coefficient for the bed roughness in MIKE 21 model, $k_s$ is roughness height.

**2.2 Salinity module**

The fundamental salinity equation is:

$$\frac{\partial h\overline{s}}{\partial t} + \frac{\partial h\overline{u}\,\overline{s}}{\partial x} + \frac{\partial h\overline{v}\,\overline{s}}{\partial y} = hF_s + hs_sS \tag{4}$$

[revised manuscript text omitted]